# A Clutter Suppression Method Based on LSTM Network for Ground Penetrating Radar

**Jianrong Geng [1], Juan He [1], Hongxia Ye [1,2,*] and Bin Zhan [3]**

1   Key Laboratory for Information Science of Electromagnetic Waves (MoE), Fudan University, Shanghai 200433, China; 20210720216@fudan.edu.cn (J.G.); 19210720097@fudan.edu.cn (J.H.)
2   State Key Laboratory of Lunar and Planetary Sciences, Macau University of Science and Technology, Avenida WaiLong, Taipa, Macau 999078, China
3   NVIDIA Technology Shanghai Co., Ltd., Shanghai 201210, China; bzhan@nvidia.com
*   Correspondence: yehongxia@fudan.edu.cn

**Abstract:** It is critical to estimate and eliminate the wavelets of ground penetrating radar (GPR), so as to optimally compensate the energy attenuation and phase distortion. This paper presents a new wavelet extraction method based on a two-layer Long Short-Term Memory (LSTM) network. It only uses several random A-scan echoes (i.e., single channel detection echo sequence) to accurately predict the wavelet of any scene. The layered detection scenes with objects buried in different region are set for the 3D Finite-Difference Time-Domain simulator to generate radar echoes as a dataset. Additionally, the simulation echoes of different scenes are used to test the performance of the neural network. Multiple experiments indicate that the trained network can directly predict the wavelets quickly and accurately, although the simulation environment becomes quite different. Moreover, the measured data collected by the Qingdao Radio Research Institute radar and the unmanned aerial vehicle ground penetrating radar are used for test. The predicted wavelets can perfectly offset the original data. Therefore, the presented LSTM network can effectively predict the wavelets and their tailing oscillations for different detection scenes. The LSTM network has obvious advantages compared with other wavelet extraction methods in practical engineering.

**Keywords:** ground penetrating radar; long short-term memory network; wavelet extraction

## 1. Introduction

Ground penetrating radar (GPR) is an electromagnetic (EM) imaging device that uses the reflection and scattering characteristics of EM waves in discontinuous media to achieve non-destructive detection [1]. It has been used in many engineering detection fields such as ground ice detection [2], underground pipeline detection, and criminal investigations [3]. Recently, space exploration probes, such as Lunar and Mars exploration, are also equipped with GPR equipment for geological stratification studies. However, due to the complex underground environment, the radar echo always contains clutter such as multiple waves, antenna coupled waves, and reflected waves from other non-detection targets, which seriously obscure the signal of the buried targets and bring great difficulty to the interpretation of GPR data.

Many scholars have conducted in-depth research into methods of clutter suppression. Some research designed an appropriate antenna system to enhance the echo of a buried object and reduce the background noise. For example, Liu et al. [4] developed a hybrid dual-polarization GPR system with one circularly polarized transmit antenna and two linearly polarized receive antennas to improve detection and estimation of slender tubular targets. Others are based on signal processing algorithms. For example, the reference wave method averages a few A-scans echoes without target information to find the wavelets of the detection scene, and then subtracts the mean from the original data [5]. The background matrix subtraction (BMS) method is an improvement of the average cancellation

method [6,7]. The background noise matrix is obtained through a series of sliding windows, sample exclusion, weighting and iteration, then it is removed from the original data to suppress the clutter. However, these methods rely heavily on prior information about the detection scene, and it is often difficult to know in advance which echoes do not contain the target signal in actual detection.

Limited by wavelet extraction, the multi-scale filtering method was developed. Bao [8] proposed a noise attenuation method based on curvelet transform. He extracted the background noise according to the distribution characteristics in the curvelet domain. Kumlu [9] designed a multiscale directional bilateral filter (MDBF), which can flexibly extract the directional details corresponding to different geometric structures from the original data. Then, the inverse MDBF is applied to reconstruct the image of targets. In addition, there are some subspace projection techniques, such as morphological component analysis (MCA) [10], Singular Value Decomposition (SVD) [11], and Principal Component Analysis (PCA) [12]. These methods project the detection signals into different subspaces and remove the noise subspace to reconstruct the clutter-free signal. However, when the complex underground environment leads to serious signal distortion or strong clutter, the wavelets will be spread in multiple principal components, which causes difficulty in determining the truncation of eigenvalues. In addition, a sparse blind deconvolution (SBD) method has recently been proposed for the GPR data process [13]. However, it would likely be more effective to estimate the optimum wavelet for each individual transmitter location separately, rather than for the whole data set. Therefore, the existing methods can only be used in specific detection environments, not suitable for actual engineering wavelet extraction.

This paper designs an LSTM neural network to directly predict the wavelet from the GPR A-scan echoes. Firstly, the possibility of the neural network to extract the wavelet is analyzed based on the characteristics of GPR signals and the neural network. Then, some A-scan data generated with a 3D-FDTD simulator are used to train a two-layer LSTM network. Some simulated and measured GPR A-scan echoes are used to test the performance of the trained network model. The results show that this extraction method can be applied to different detection environments without any prior information. It can avoid the heavy marking tasks in large-scale detection areas. It can also solve the calibration in special detection environments.

## 2. Detection Principle and Data Characteristics of GPR Radar

As shown in Figure 1a, the GPR detection trolley is used to detect the buried object in a sand bunker. The trolley moves at a constant speed along a straight line to collect 512 channels of A-Scan data; each has 1024 samples. Generally, the convolution of GPR wavelet and reflection sequence constitutes the echo of GPR detection system. The radar first receives the strong reflection signal of the upper surface. When the transmitted pulse encounters the target in the underground, there is a reflected signal of the target, which is relatively weak. Figure 1b is the time stacking diagram of the first 512 sampling points. It can be seen that the strong wavelet submerges the signal of target. Moreover, the wavelets at the adjacent monitoring points on the same survey line in the same detection area are very similar.

Three correlation coefficients of Pearson's linear correlation coefficient [14], Kendall's tau coefficient [15], and Spearman's rho [16] are used to evaluate the relevance of adjacent A-Scan echoes. The correlation coefficient greater than zero indicates that the two groups of data are positively correlated. On the contrary, a correlation coefficient less than zero indicates that they are negatively correlated. Moreover, the greater absolute value indicates the stronger correlation. Three correlation coefficients of the A-Scan echoes in Figure 1 are shown in Figure 2. The mean values are 1, 0.9998, and 0.9894, respectively, which proves the strong correlation between adjacent echoes. Several isolated valley points on the curves correspond to the buried target, especially on the line of Kendall's tau. This strong correlation in time series could be well explored by neural networks with memory.

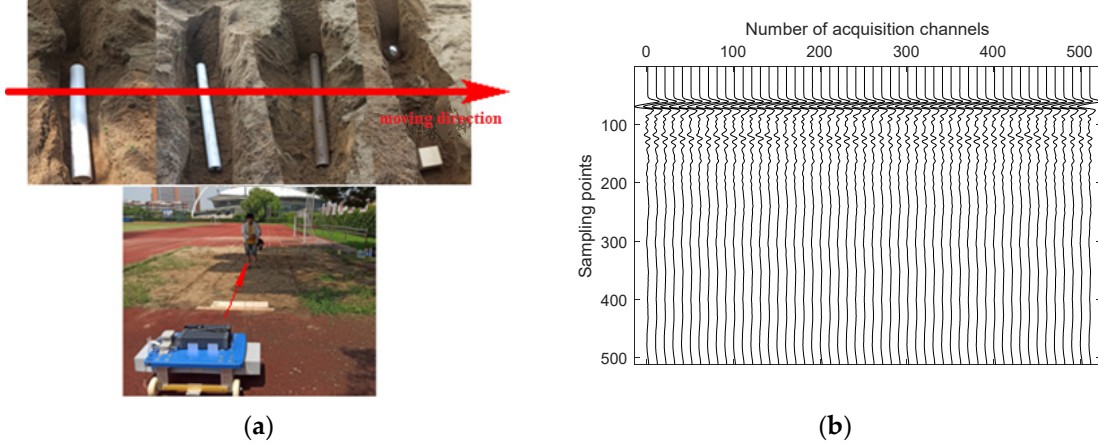

**Figure 1.** Detection of buried targets by ground penetrating radar. (**a**) Schematic diagram of GPR measurement. (**b**) Time accumulation diagram.

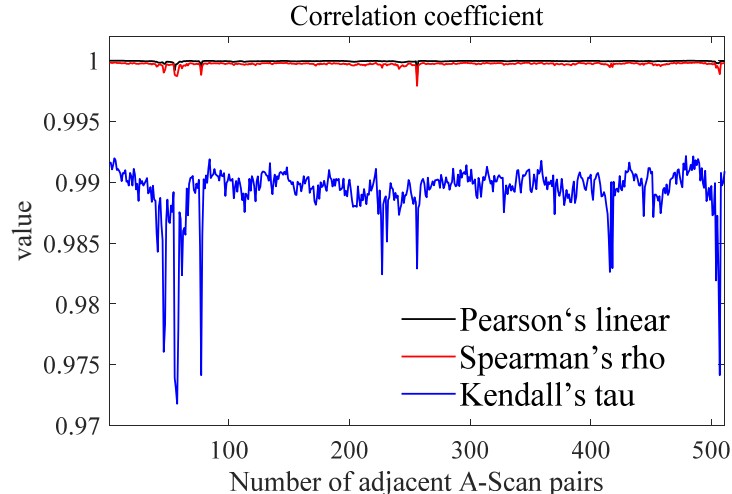

**Figure 2.** Correlation coefficients of the adjacent A-Scans in Figure 1.

## 3. Two Layers LSTM Network for Wavelet Prediction

### 3.1. The Structure of Two-Layer LSTM Model Network

Recurrent neural networks (RNNs) are commonly applied to explore relations in sequential data. The special network structure can selectively store the past information and use them together with current input to speculate the future information. Long Short-Term Memory (LSTM) is a variation of RNN with the capability to prevent gradients decaying or exploding. It can fully explore the non-linear relationship between variables and process complex long-term time series dynamic information [17].

Figure 3 shows the cell structure of LSTM network, which has a new memory unit $R_t$ and three control gates, namely input gate ①, forget gate ②, and output gate ③. The computing flow is expressed as the following equations [18].

$$f_t = \sigma(w_F[y_{t-1}, x_t] + b_F) \tag{1}$$

$$i_t = \sigma(w_I[y_{t-1}, x_t] + b_I) \tag{2}$$

$$R\prime_t = tanh(w_R[y_{t-1}, x_t] + b_R) \tag{3}$$

$$o_t = \sigma(w_O[y_{t-1}, x_t] + b_O) \tag{4}$$

$$R_t = f_t * R_{t-1} + i_t * R\prime_t \tag{5}$$

$$y_t = o_t * tanh(R_t) \tag{6}$$

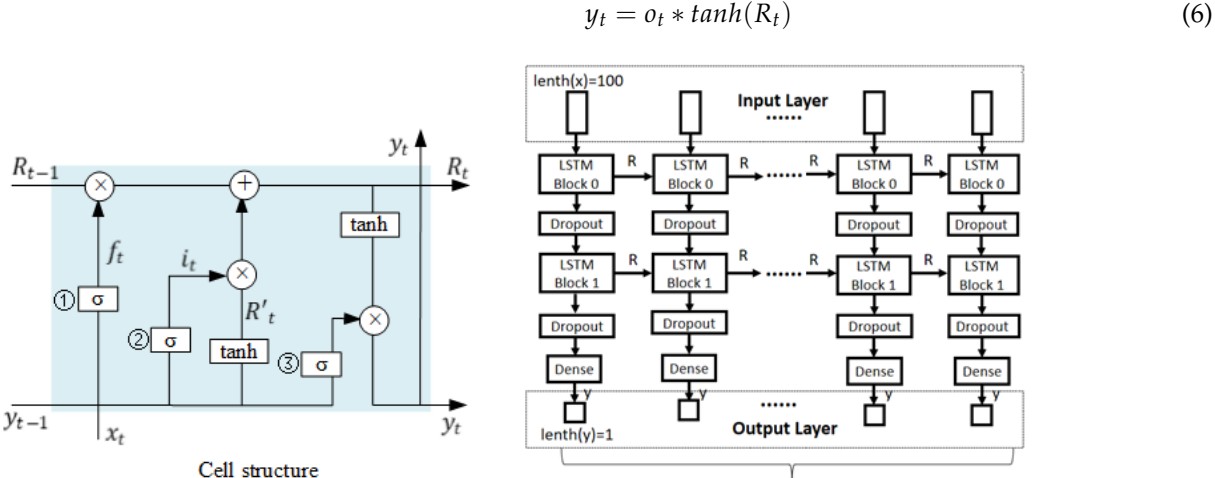

**Figure 3.** The structure of two-layer LSTM model network.

Here, $y_t$ is the output and $x_t$ is the input. $\sigma$ is the sigmoid function. *tanh* is the hyperbolic tangent function. $w$ and $b$ are the weights and biases, respectively. The subscripts *I*, *F*, *O*, and *R* represent the input gate, forget gate, output gate and memory unit, respectively. The symbol "$*$" means convolution. As is well known, the wavelet sequence of a given GPR system and detection environment is always stable. The characteristics of LSTM network makes it possible to extract the wavelet sequences from the GPR echoes. In the following, multiple A-scan echoes on one survey line are connected end to end to form a longer sequence for wavelet prediction.

The network structure contains two-layer LSTM, two-layer Dropout, and one-layer Dense. The dropout probability of 0.2 is set to prevent overfitting. The timestep is 100, batch size is 200, epoch is 200, and learning rate is $1 \times 10^{-6}$. The input length of each timestep is 100, and the output length is 1. If the loss does not decrease within 10 epochs, the learning rate is dynamically adjusted to be 0.1 times that of before.

*3.2. Network Training*

As shown in Figure 4a, three-layer infinite medium is set up to simulate air, cement, and limestone from top to bottom. The relative permittivity of cement and limestone are $\varepsilon_{r2} = 6$ and $\varepsilon_{r3} = 9$, respectively. The cement layer thickness is $d_1 = 0.4$ m. The top air layer and the bottom limestone layer are both infinite. A PEC cube object with a 0.2 m edge length is partially buried between the cement layer and the limestone layer. The depth from the top of the cube to the ground is $d = 0.3$ m. The distance between the transmitting and receiving antennas is $L = 0.4$ m, and the height above the ground is $h = 0.4$ m. The two antennas are vertically located on both sides of the survey line and move synchronously along the survey line of the red arrow. The signal source is a ricker wave with center frequency of 200 MHz, time sampling interval of 0.0385 ns. The spatial sampling interval along the survey line is 0.04 m. A 3D-FDTD simulation tool [19] is used to generate the A-scan data. In total, 95 channels of A-Scan echoes, each with 780 samples, are accumulated along survey line to form the B-Scan image, as shown in Figure 4b. In order to eliminate the signal similarity of dense sampling along the survey line, we selected 30 channels of data at equal intervals as the data set. From these data, 18 channels of A-scan echoes are randomly selected for the training set, 6 channels are for the validation set, and another 6 channels are for the testing set.

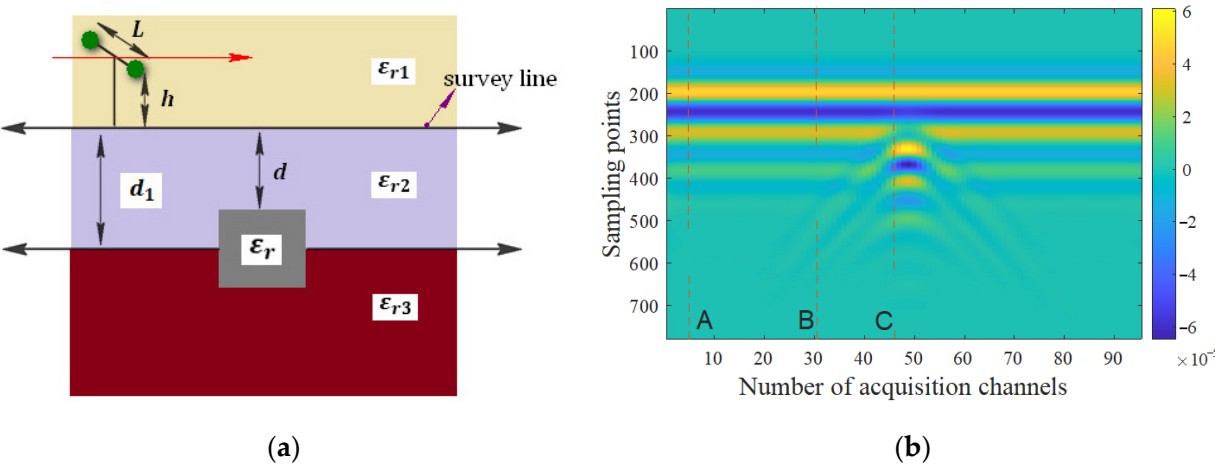

**(a)**                                                    **(b)**

**Figure 4.** The original simulation model and the corresponding B-scan image. (**a**) Schematic diagram of GPR measurement. (**b**) Time accumulation B-scan image.

The 18 channels of A-scan echoes of the training set are randomly connected end-to end to construct a longer sequence with $780 \times 18 = 14{,}040$ samples as the input sequence. In each step of network training, the input length is 100, which means the first 100 samples of the input sequence are used to predict the next sample. The time window of the input data moves backward step by step. The network uses the mean square error (MSE) as the loss function.

Figure 5 shows the loss during the training process. The loss of the one-layer network decreases slowly, and the final loss is much greater than the loss of the two-layer network. The loss of the three-layer network drops quickly, but it appears a slight over-fitting has resulted in poor prediction results. The two-layer network converges after the 180 steps, with the loss less than 0.001. Figure 6 compares the predicted wavelet. It can be seen that the red line predicted by the two-layer network is most similar to the ideal wavelet, which is the echo when there is no target buried in the ground. So, the two-layer LSTM network is superior in training time and performance and will be used for wavelet prediction in the following.

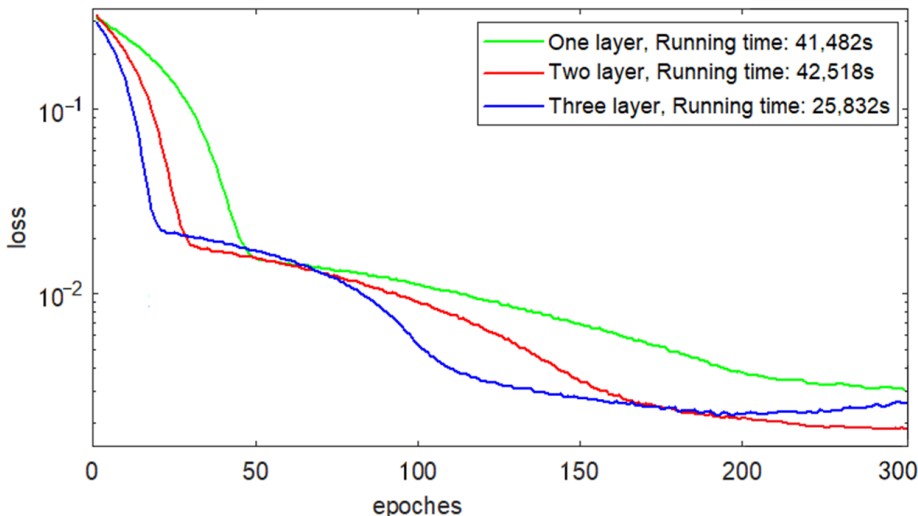

**Figure 5.** Loss function of network training process.

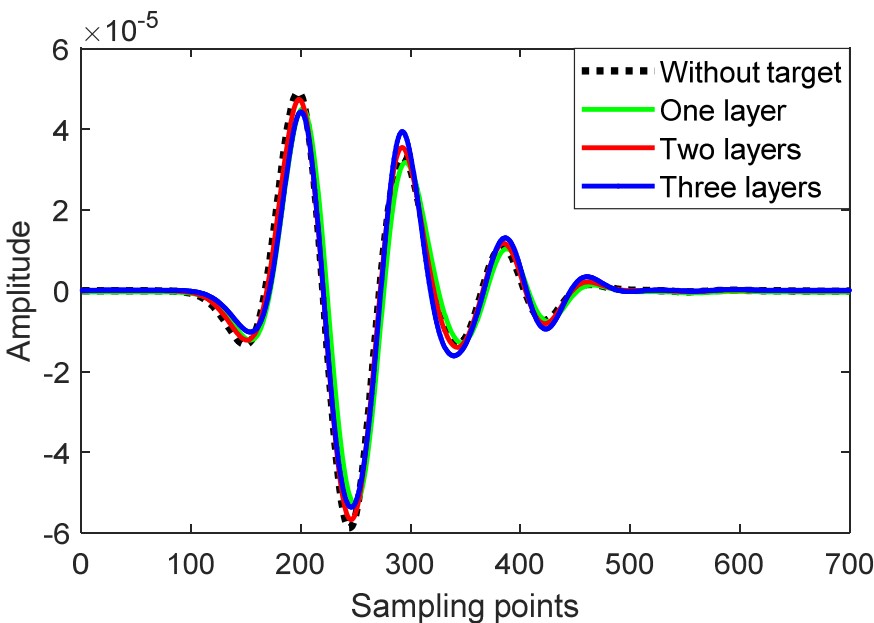

**Figure 6.** Wavelets predicted by different network layers.

### 3.3. Network Testing

From the test set, we randomly selected two channels to connect into a sequence with 780 × 2 = 1560 samples as the network input for the prediction of wavelet. For example, the 5th, 32nd, and 48th channels of A-Scan data (corresponding to A, B, and C in Figure 4b, respectively) are used to evaluate the wavelet prediction ability of the trained network. Position A is far away from the target, and it contains less target signal. This channel of A-scan data is used as the ground truth of the wavelet. Position B is close to the target, and the corresponding A-scan echo contains some target signals. As shown in Figure 7, the predicted wavelet of green line is consistent with the ground truth of black dot line. The enlarged view in the upper right corner shows that the predicted wavelet amplitude is non-zero after the 510th sampling point, due to the reflected signal of the underground layer.

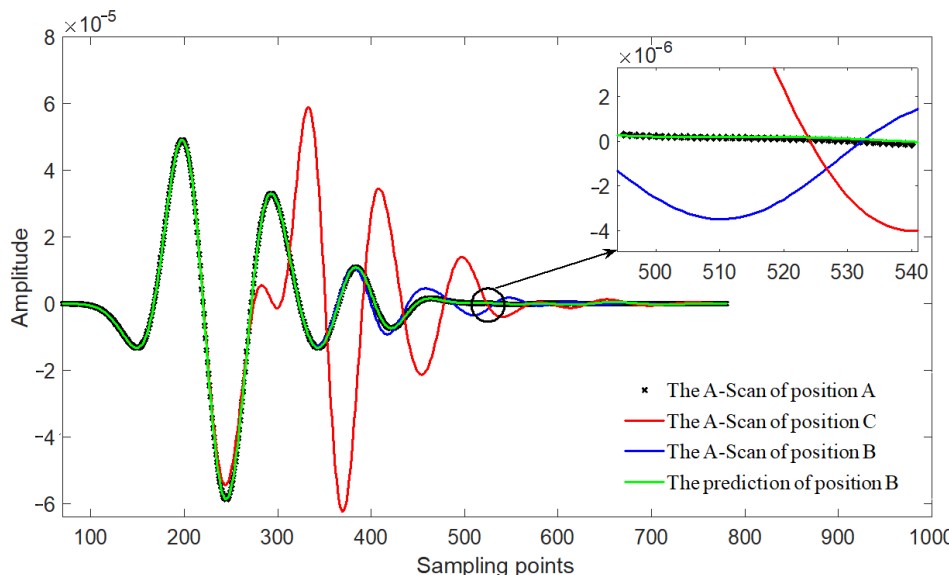

**Figure 7.** Comparison of prediction results with original A-Scan data.

Another interesting observation is the wavelet prediction with the A-scan data at position C, which is just above the PEC target. The strong reflected signal of the metal is much larger than the wavelet of the background. At this time, the network cannot ignore the very strong target interference, resulting in the failure of wavelet prediction. Therefore, for the wavelet prediction of the actual measured data, it is necessary to remove the echo containing strong target signal.

The effect of wavelet removal and deconvolution strongly depends on the quality of the wavelet. If the extracted wavelet is incorrect, the direct wave cannot be offset with the original data, and interference is also induced. The inaccurate wavelet tailing will affect the resolution of deep detection. In the following, the results of several methods of wavelet removal and deconvolution are compared to evaluate the accuracy of wavelet.

Figure 8 shows the results of wavelet removal with different wavelet extraction methods. The above two figures still contain residual direct waves and the layered interference. Although the reference wave method removes the direct wave cleanly, the layered interference is still obvious. The removal of the wavelet predicted by the above LSTM network can effectively remove the direct wave and the layered interference at the same time and improve the signal-to-noise ratio and resolution.

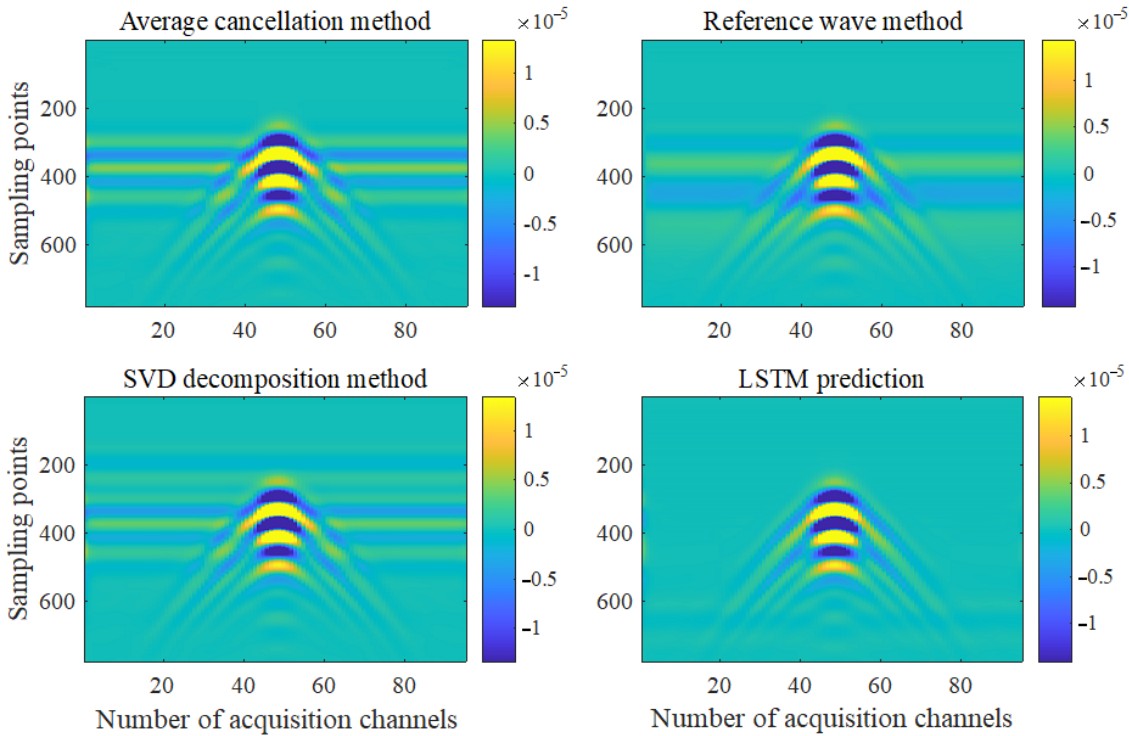

**Figure 8.** Wavelet removal extracted by four methods.

Figure 9 shows the deconvolution results of the wavelets. It can be seen that the deconvolution with the LSTM-predicted wavelet can effectively compress the wavelet tailing and the layered signal so that the signal of the deep target is highlighted.

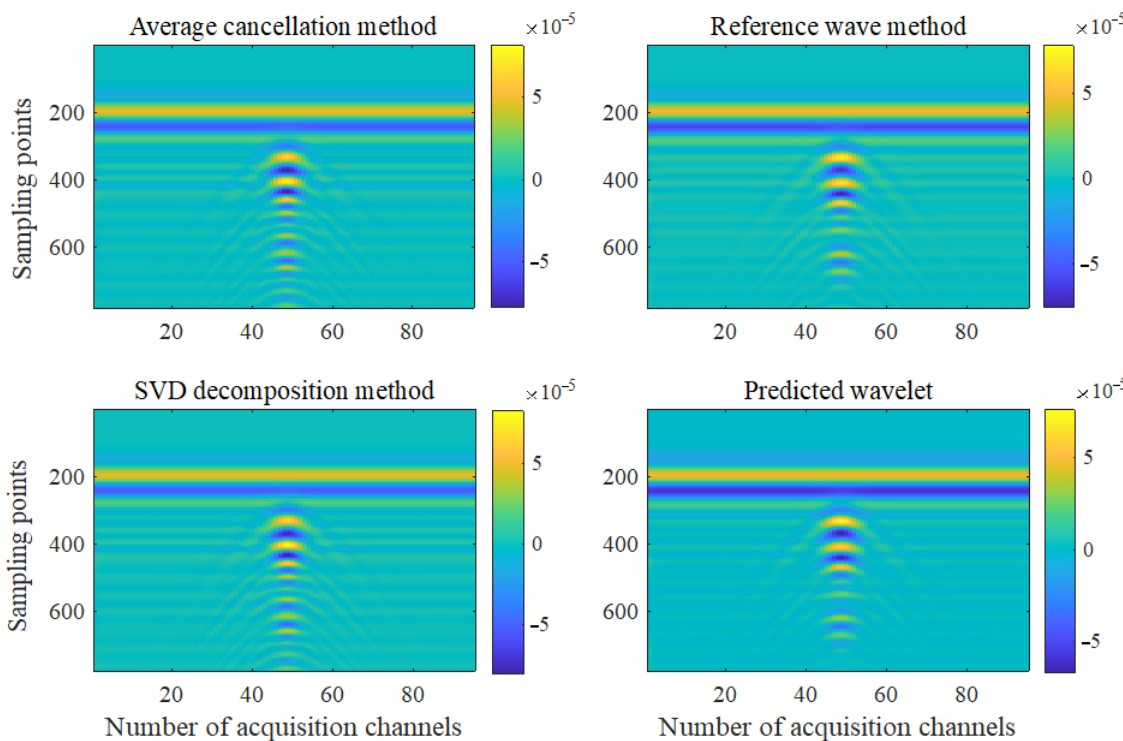

**Figure 9.** Comparison of deconvolution processing.

## 4. Network Generalization Experiments

### 4.1. Other Simulation Scene

According to the convolution echo model of the GPR system, regardless of environmental noise and other interference, the GPR wavelet corresponds to the collected data without a buried target. However, the engineering GPR detection will encounter the following problems: (1) The underground layered structure is diverse. The number of layers and layer thickness in mountainous, rural, and urban areas are very different; (2) The shape and material of the buried targets are different; (3) The background medium of the detection environment is complex, such as clay, sand, gravel, etc. Their dielectric constants are different; and (4) Different radar systems may use different source waveforms. For example, the LTD series GPR of China Research Institute of Radio wave Propagation (CRIRP) uses Ricker, while the lunar or Mars rover GPR often uses the chirp signal. In the following, we set up two groups of simulation experiments to evaluate the feasibility of the above LSTM network method for wavelet prediction in different detection scenes. These simulation scenes are derived from Figure 4 by changing the scene parameters to create eight new models. The 3D-FDTD simulator is used to generate the A-scan and B-scan data. In total, 95 channels of A-Scan echoes, with 780 samples for each channel, are simulated for each new model.

### 4.1.1. Different Layered Structure

In the first group of experiments, we change the layered structure. We firstly add another layer at different depths to generate two new models, as shown in Figure 10. The network trained above with the original model in Figure 4 does not need to be trained again. We select three channels of A-scan data, corresponding to positions A, B, and C, to test the generalization ability of the LSTM network trained above. Position A is far away from the target, so the corresponding A-scan echo, shown as the black line in Figure 10, is used as the ground truth of the wavelet for the new scene. The yellow line and the green dotted line are the predicted wavelet with the A-scan echo at position B and position C, respectively. It can be seen that the wavelet of the new scene contains a new signal peak at

the tail of the echo, which is consistent to the new layer position. Therefore, the proposed network can predict the response of the new layer structure.

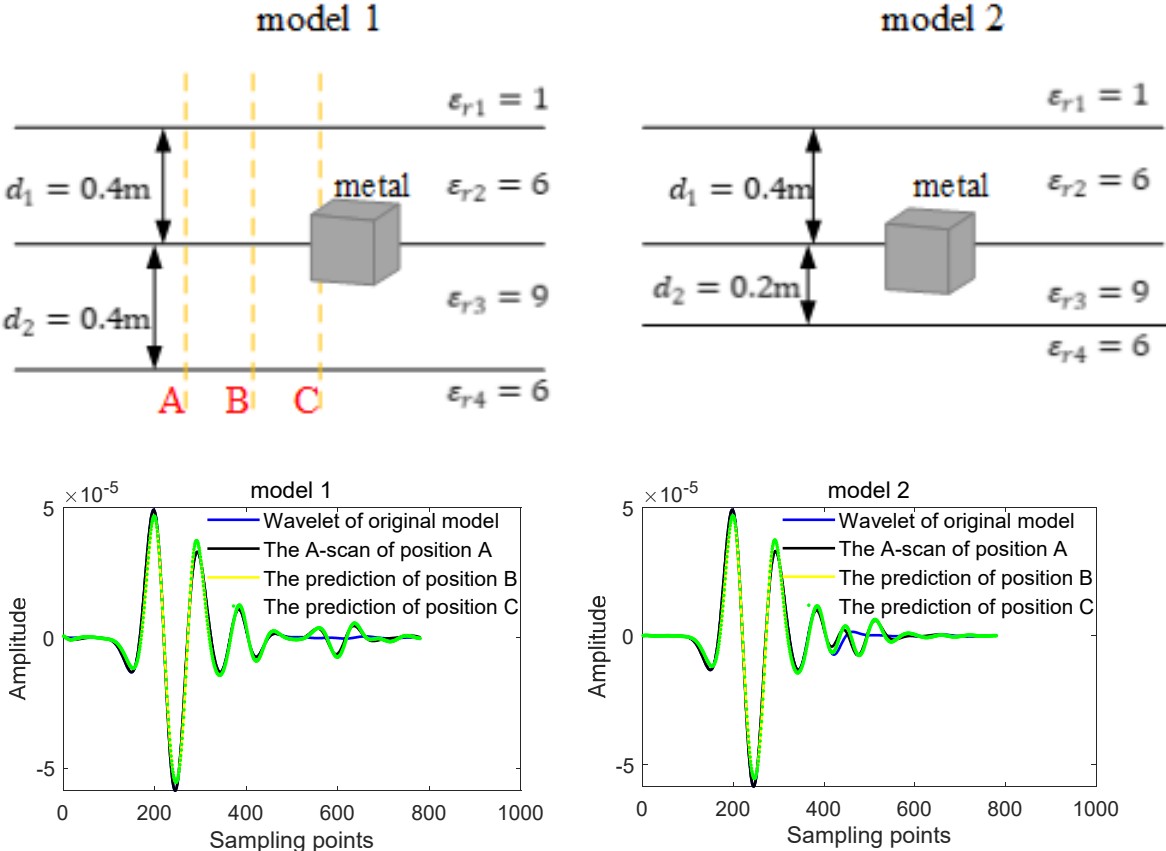

**Figure 10.** Wavelet prediction results of derived models 1–2.

As shown in Figure 11, model 3 only changes the thickness $d_1$. According to the layered geometry, the position of the second layer signal will change from the 387th sampling point to the 465th sampling point. Model 4 further changes the dielectric constants of the two layers, and the position of the second layer signal is moved to the 417th sampling point. The LSTM network trained above with the original model is used to predict the wavelet for the new model directly. The prediction results correctly indicate these layered geometries. Therefore, the proposed network can capture the layer thickness and background medium to make accurate predictions quickly.

The predicted wavelets of the new models are used for background elimination. Figure 12 indicates that the predicted wavelets removal can highlight the target information, while the B-scan obtained with other methods still has some layered signal residues. Therefore, the LSTM wavelet prediction network model is suitable for the layered medium model of different layers and thicknesses.

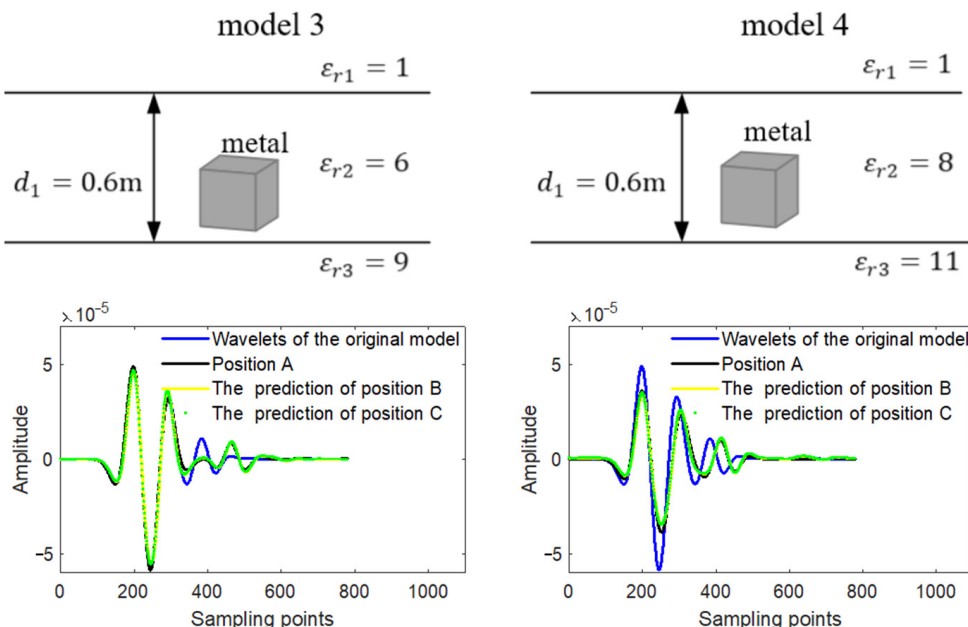

**Figure 11.** Wavelet prediction of derived models.

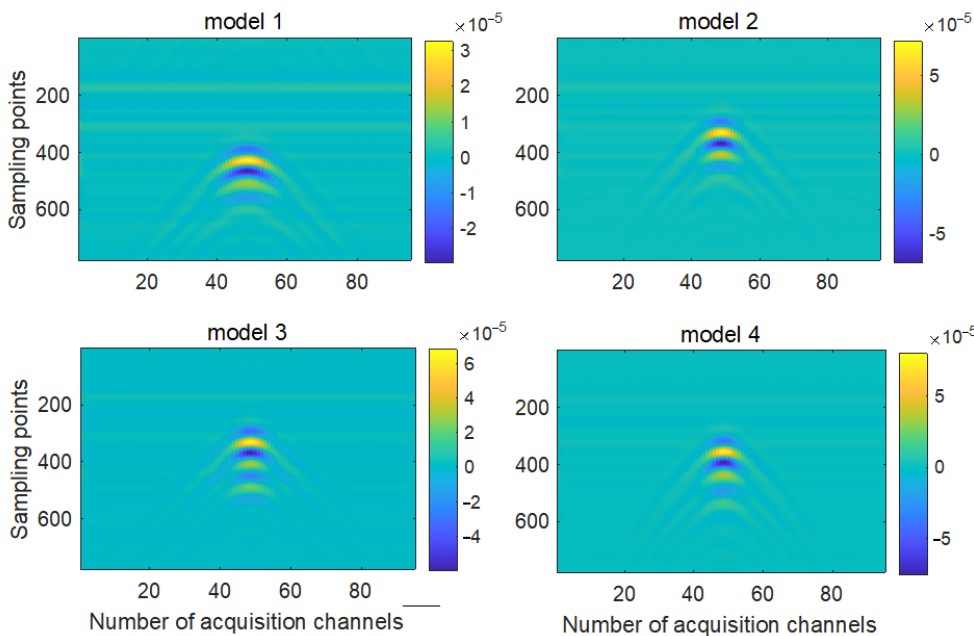

**Figure 12.** Predicted wavelet removal results of models 1–4.

4.1.2. Different Radar System and Underground Medium

In the second group of experiments, we consider the change of multiple parameters, including source waveform, background permittivity, target shape, and depth. The layered geometry of the background is the same as that of model 1. The original cubic PEC target is replaced by an infinite air cylinder with radius R. The simulation parameters are listed in Table 1.

Figure 13 shows the prediction results of models 5–8. The blue line is the wavelet of the original model in Figure 4. The black line is the A-Scan without the target signal, and it is the ground truth of the wavelet for the new model, which is quite different from the original model. The network is directly used to predict the wavelet for the new models, without retraining. The red line is the predicted wavelet with the A-can echo at position

B of the derived model. Although multiple parameters, such as the layered geometry, medium parameters, target information, and excitation waveform, are all different from the original model in Figure 4, the LSTM network still can predict the accurate wavelets.

**Table 1.** Simulation parameters of derivative models 5–8.

| Model | Source | $\varepsilon_{r2}$ | $\varepsilon_{r3}$ | $d$ (m) | $R$ (m) |
|-------|--------|--------|--------|-------|-------|
| 5 | Ricker | 5 | 8 | 0.3 | 0.2 |
| 6 | Ricker | 10 | 13 | 0.3 | 0.2 |
| 7 | Ricker | 9 | 1 | 0.24 | 0.16 |
| 8 | Gaussian dot norm | 5 | 7 | 0.2 | 0.2 |

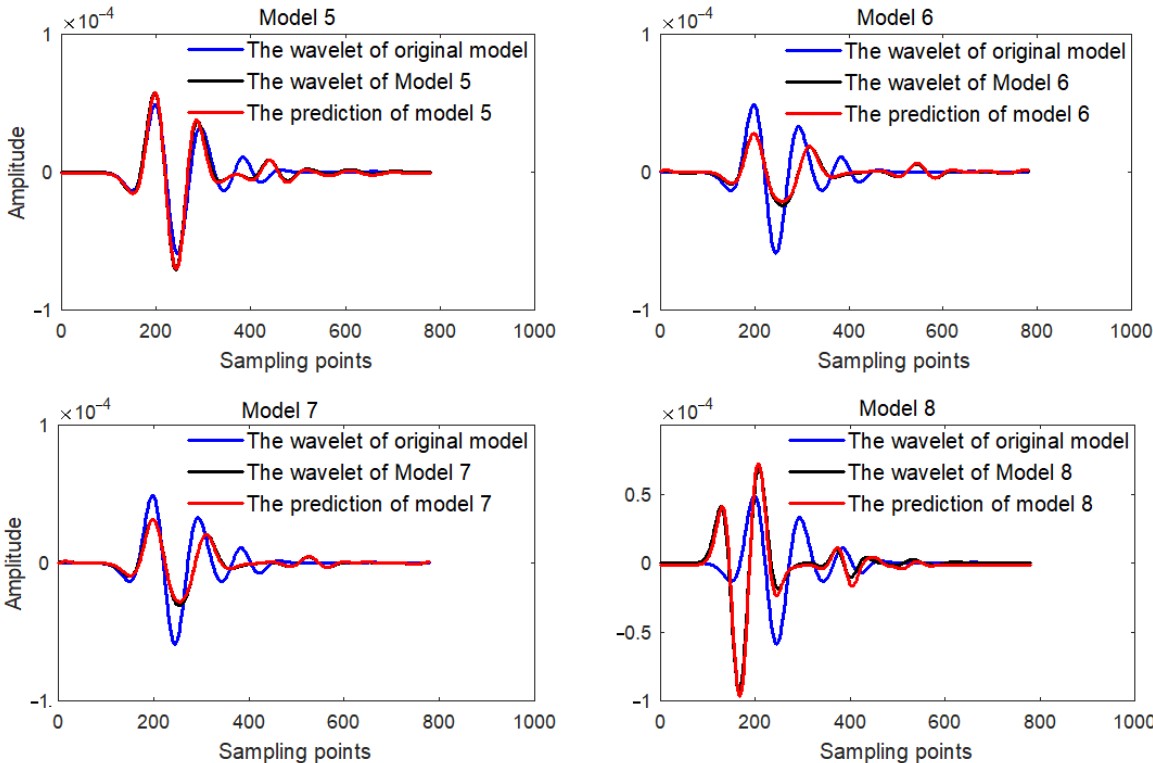

**Figure 13.** The predicted wavelet of derived models 5–8.

*4.2. Wavelet Prediction of Measured Data*

In order to explore the practicability of this method, a group of measured data are used to test the performance of the above method for wavelet extraction. Several landmines are buried in red clay and clay scenes for experiments, and the LTD series GPR of China Research Institute of Radio wave Propagation (CRIRP) is used to collect echo data. Red clay is a soil with high water content formed by carbonate weathering and has a large relative permittivity $\varepsilon_r = 12$. The water content of the clay is low, and the relative permittivity $\varepsilon_r = 12$ is low. The parameters of red clay and clay are listed in Table 2. There are 893 and 692 channels of A-scan data from red clay and clay scenes, respectively, each with 1024 sampling points, as shown in Figure 14. Different background media cause different attenuation of EM waves in two scenarios. Therefore, the echo data are significantly different.

**Table 2.** Parameters of red clay and clay.

| Soil Type | Water Content | $\varepsilon_r$ |
|-----------|---------------|------|
| Red clay | 45% | 12 |
| Clay | 20% | 4 |

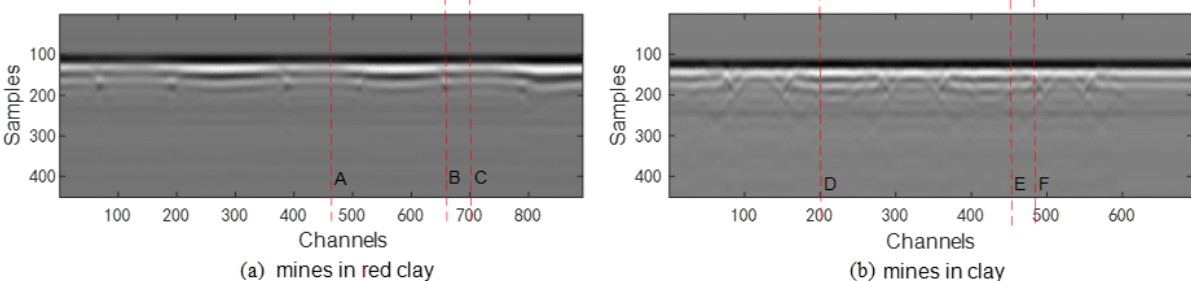

**Figure 14.** The detection data of mines in red clay and clay. (The channel numbers of A–F are 461, 659, 701, 200, 449, and 483, respectively).

The antenna parameters of the radar system are different from the simulation model. The radar system parameters often affect the wavelet prediction network. If you want to use the network trained with the simulation data to predict the wavelet of the actual measurement data, you must accurately set the antenna parameters in the simulation model. However, it is difficult to measure the system parameters of the actual detection equipment accurately. Moreover, system components, noise, and the internal interference is difficult to be reproduced ideally. According to the learning characteristics of LSTM network for sequence correlation, we try to train the network directly with the measured data.

A total of 38 channels of A-scan data of the red clay detection are randomly selected to splice in sequence as the training set. The training process is the same as above and ends when the loss is less than 0.0001. Then, another 12 different channels of A-scan data are randomly selected to predict the wavelet, as shown in Figure 15. It can be seen that the network can well predict the wavelet whether the echo contains the target signal or not. The strong signal at the 200-th track of the blue line corresponds to the reflected signal from the mine. The LSTM network can separate the wavelet from the target signal and environmental interference. Then, the trained network with red clay data is directly used to predict the wavelet of the clay scene, as shown in Figure 15b. The predicted wavelet is close to the A-Scan without the target signal, which is the ground truth of the wavelet for the clay background.

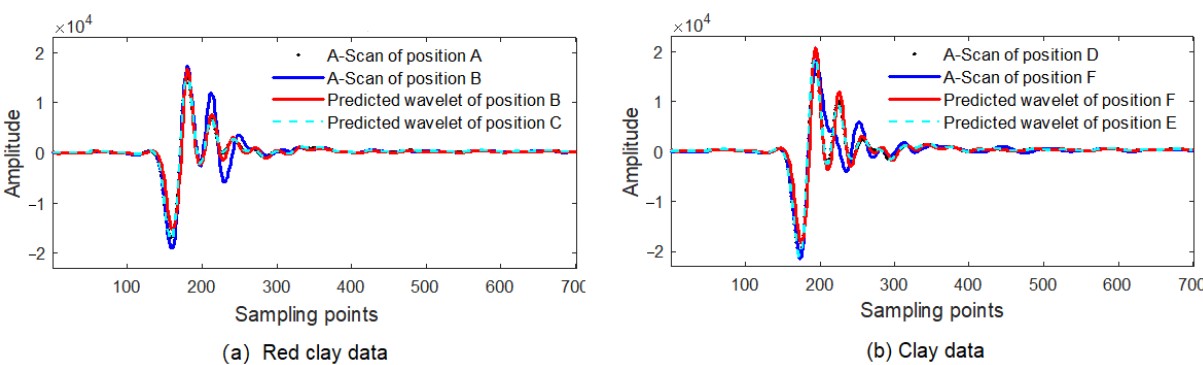

**Figure 15.** Wavelet prediction of the detection data in Figure 14.

## 5. Conclusions

This paper presented a wavelet prediction method based on the LSTM network. This method takes advantage of the strong correlation of the GPR signals to construct quasi-periodic input signal by splicing the data head and tail. The network is trained to learn the commonality of the input data, while ignoring the random interference and predicting a smooth wavelet.

Several groups of experiments (corresponding to different excitation source waveform, target location, shape, material, background geometry, etc.) show the generalization ability of the network for different detection scenarios. As long as the antenna system parameters

remain unchanged, the network trained with the simulation data of one arbitrary scene can be used to predict the wavelet of many different scenes. In order to expand the applicability of the network to different media, the network is optimized by expanding the range of dielectric parameters of the simulation model. For the application of actual detection data, we directly use a small amount of measured data for network training, and then use the trained network for detection data in other different scenes. Compared with the wavelet extracted by SVD, the predicted wavelet has obvious advantages in the integrity and adaptability of the detection area, and the method can be used in large-scale underground exploration projects such as pipeline detection under urban roads, defect detection inside a tunnel, and so on.

This method has many advantages, as follows:

1. This method does not rely on prior knowledge and can effectively extract the wavelets of different scenes.
2. There is no need for artificial marking during network training. The A-Scan echo can be directly used as training data, and the input data is easy to obtain.
3. The trained network has good generalization ability and can solve many practical problems, such as heavy marking of a large-scale detection area, the inability to label special detection environments, and poor processing results caused by inaccurate calibration.

The proposed neural network method has very strong generalization ability for wavelet prediction of the same antenna system. However, if the antenna system is changed, the network must be retrained. This issue needs to be further studied in the future. The probable approach is to conduct a large number of network training experiments by controlling a single variable or multiple variables, such as antenna system parameters or the environment parameters. The large number of detection data will make the network learn its internal connections. Moreover, we will also try to use the Transform network structure to predict.

**Author Contributions:** Conceptualization and Supervision, H.Y.; Writing—Original draft, J.H.; Writing—Review and editing and Software, J.G.; and Methodology and Validation, B.Z. All authors have read and agreed to the published version of the manuscript.

**Funding:** This research was funded by in part by the National Key RD Program of China 2021YFB3900100 and in part by the National Natural Science Foundation of China (Grant No. 61871424).

**Institutional Review Board Statement:** Not applicable.

**Informed Consent Statement:** Not applicable.

**Data Availability Statement:** Not applicable.

**Conflicts of Interest:** The authors declare no conflict of interest.

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
