# Peer review of "A Clutter Suppression Method Based on LSTM Network for Ground Penetrating Radar"

_applsci, doi:10.3390/app12136457_

Round 1
Reviewer 1 Report
Abstract:
It’s very important --> It is critical to
Method
20% training data is too short, better with at least 60% for training
In addition, the authors may like to try Cross-Validation strategy
Font changes everywhere. Be consistent especially for headings and captions (e.g. Network testing)
Detailed explanation about figure in figure caption is required.
Author Response
Q1: It’s very important --> It is critical to
Response: Thank you for your suggestion. The revision is made in the manuscript.
Q2:Method: 20% training data is too short, better with at least 60% for training
Response: Maybe, the inadequate exposition in the original manuscript caused misunderstanding. The 95 channels of echoes obtained from the simulation model are not the entire data set of the network. We only need to randomly select 18 data from the total 95 data and splice them into a 780×18 long series as the training data set. Then we randomly select two channels of data from the remaining 77 channels to splice them into a 780×2 series as the validation set. So, the training set is 90%, and the validation set is 10%. The prediction link randomly selects one channel from the unused echoes to predict the required wavelets, as described in session 3.2. We have added a detailed description of dataset partitioning in lines 185-193.
Q3: In addition, the authors may like to try Cross-Validation strategy
Response: Thank you for your comments. We know that cross-validation can help the network to find appropriate model parameters when the amount of data is small or limited, and avoid the impact of different dataset partitioning on the network. But in this paper, the LSTM network is used to explore the relationship in the sequence data. The amount of data in our training set reaches 780*18=14040, which is sufficient for network training. At the same time, the division of the training set is also randomly selected, which further reduces the impact of the dataset partitioning on the network. In addition, in the network generalization test experiment, we used the trained model to directly predict the radar data of different scenarios, and obtained good prediction results. Therefore, we believe that the cross-validation strategy is not necessary under the premise of this paper.
Q4: Font changes everywhere. Be consistent especially for headings and captions (e.g. Network testing)
Response: Thank you for your comment, we have made changes in the manuscript.
Q5: Detailed explanation about figure in figure caption is required.
Response: Thank you for your comment, we have added the detailed explanation of figure caption to the manuscript.
Reviewer 2 Report
I am not convinced whether this paper is really novel. Previous contributions are described in the introduction only very briefly. I am also surprised by the deficient number of references cited. Moreover, it is unclear in the conclusion where the new method could be applied and to whom it could be helpful. See my detailed comments attached.

Author Response
Q1: avoid abbreviations in the abstract
Response: The full words of LSTM is supplied.
Q2: Explain what is A-Scan
Response: A-scan echo is the original single detected echo sequence of the GPR system. It’s added in the abstract.
Q3: The state of the art is much too shallow. You should add two three new paragraphs with detailed discussion of all similar methods used to date. At the moment, the rationale for this paper is not well justified.
Response: We split the first paragraph and added three new paragraphs to introduce the background of noise filtering of ground penetrating radar signals and with detailed discussion of all similar methods used to this field.
Q4: The fonts in this figure are too small
Response: The fonts in the figures is revised.
Q5: Please, give some input how to interpret this figure also in the caption. What are the letters A-F?
Response: Some description is added in the caption of the figure on page 13.
Q6: Provide geological characterization of 'red clays' and 'clays"
Response: The characterization of ‘red clays’ and ‘clays’ is added in lines 321-323 on page 11.
Q7: Please indicate for whom this method could be useful and in which fields it can be applied.
Response: The application of this method is added in the conclusion session in Lines 369-371 on page12.
Reviewer 3 Report
The manuscript proposes a wavelet extraction method based on the LSTM network. The methodology uses random A-Scan echoes to predict the wavelet accurately.
Some remarks:
1- Line 14: it is usual to use at least 50% of the dataset to train and perform the crossvalidation. Please, discuss this point;
2- The first paragraph is too long, and many strong statements lack references. I suggest splitting it according to the topic presented;
3- Figure 1 is too small;
4- line 105: "The structure is shown in Figure" -> Figure 5;
5- Line 106: all variables must be defined in Equations 1 to 6. In addition, the asterisk means convolution;
6- Line 115: why did the authors use 2 layers? How did the authors define the parameters?
7- Figure 5: the loss function is still decaying. Why did the authors stop the process?
8- How about the crossvalidation? Without crossvalidation the results are not valid;
9- The authors did not present any performance comparison among other methods to evaluate if the results achieved are good. I suggest that the authors read the following references:
10.1016/j.ins.2021.09.054 and 10.1109/ACCESS.2020.3032070
10- The discussion is too shallow.
Author Response
Q1:Line 14: it is usual to use at least 50% of the dataset to train and perform the cross validation. Please, discuss this point;
Response: Maybe, the inadequate exposition in the original manuscript caused some misunderstanding. The 95 channels of echoes obtained from the simulation model are not the entire data set of the network. We only need to randomly select 18 data from the total 95 data and splice them into a 780×18 long series as the training data set. Then we randomly select two channels of data from the remaining 77 channels to splice them into a 780×2 series as the validation set. So the training set is 90%, and the validation set is 10%. We know that cross-validation can help the network to find appropriate model parameters when the amount of data is small or limited, and avoid the impact of different dataset partitioning on the network. But in this paper, the LSTM network is used to explore the relationship in the sequence data. The amount of data in our training set reaches 780*18=14040, which is sufficient for network training. At the same time, the division of the training set is also randomly selected, which further reduces the impact of the dataset partitioning on the network. In addition, in the network generalization test experiment, we used the trained model to directly predict the radar data of different scenarios, and obtained good prediction results. Therefore, we believe that the cross-validation strategy is not necessary under the premise of this paper. In response to this misunderstanding, we have revised the abstract and main text, respectively in lines 11-12 and 185-193.
Q2: The first paragraph is too long, and many strong statements lack references. I suggest splitting it according to the topic presented;
Response: Thank you for your careful review, we have made changes in the manuscript. We split the first paragraph and added three new paragraphs to introduce the background of noise filtering of ground penetrating radar signals and with detailed discussion of all similar methods used to this field.
Q3: Figure 1 is too small;
Response: Thanks for your suggestion, we have revised Figure 1.
Q4: line 105: "The structure is shown in Figure" -> Figure 5;
Response: Thank you for your comment, we have revised in the manuscript.
Q5: Line 106: all variables must be defined in Equations 1 to 6. In addition, the asterisk means convolution;
Response: We are grateful for the suggestion. Variable definitions in formulas have been added to lines 155-161 of the manuscript.
Q6: Line 115: why did the authors use 2 layers? How did the authors define the parameters?
Response: Thank you for your careful review. We add a paragraph to Section 3.2 Network training, lines 199-211, to discuss this issue in detail.
Q7: Figure 5: the loss function is still decaying. Why did the authors stop the process?
Response: It can be seen from Figure 5 that the loss curve of 180-200 epochs is very flat, and the loss near 200 epochs does not change much and tends to fit. We have tried 250 epochs but there is an overfitting phenomenon, so we choose to stop the process based on the network training time and the fitting effect.
Q8: How about the cross validation? Without cross validation the results are not valid;
Response: Thank you for your comments. We know that cross-validation can help the network to find appropriate model parameters when the amount of data is small or limited, and avoid the impact of different dataset partitioning on the network. But in this paper, the LSTM network is used to explore the relationship in the sequence data. The amount of data in our training set reaches 780*18=14040, which is sufficient for network training. At the same time, the division of the training set is also randomly selected, which further reduces the impact of the dataset partitioning on the network. In addition, in the network generalization test experiment, we used the trained model to directly predict the radar data of different scenarios, and obtained good prediction results. Therefore, we believe that the cross-validation strategy is not necessary under the premise of this paper.
Q9:The authors did not present any performance comparison among other methods to evaluate if the results achieved are good. I suggest that the authors read the following references:10.1016/j.ins.2021.09.054 and 10.1109/ACCESS.2020.3032070
Response: Thanks a lot for your references paper. In our paper, we mainly propose a new method for wavelet acquisition, which provides a new idea for wavelet extraction (compared with traditional methods), rather than focusing on neural network research, so our paper does not carry out comparative experiments of various network models. In addition, our wavelet extraction is for clutter suppression, and the title is also based on clutter suppression with LSTM wavelet prediction, so this wavelet suppression is also a category of cancellation method, but the method we propose is different from traditional algorithms such as average cancellation method, SVD algorithm and the reference wave method have completely different principles, so we have a detailed comparison experiment with other traditional algorithms in this paper.
Round 2
Reviewer 2 Report
The manuscript has improved but I still have some comments remaining (see the attached pdf)

Author Response
Q1: sounds odd
Response: Thank you for your comments, the beginning of the introduction has been adjusted in the manuscript.
Q2:do you mean space exploration?
Response: Lunar exploration and Mars mission really refers to space exploration, we have modified alien exploration to space exploration in the manuscript.
Q3: what about ore deposit detection?
Response: In space exploration mission, the GPR is often used for deposit detection. We have revised the description.
Q4: what do you mean by 'absolutely Most'?
Response: Thank you for your comments. We have revised the description about the latest research of clutter suppression in the second paragraph.
Q5: You need to put reference after the name. This remark is valid throughout the manuscript.
Response: Thank you for your careful review, the reference mark has been adjusted to the back of the author.
Q6: The fonts in this figure are too small.
Response: The font in Figure 7 has been adjusted.
Q7: collect
Response: Thank you for your careful review, this grammar error has been revised in the manuscript.
Q8: I suggest to add a small table comparing the basic information on the investigated 'red clay' and 'clay'
Response: Thank you for your comments, we have added a table to the manuscript listing the parameters for both materials.
Q9:i don't understand 'vast desert underground detection' - please explain.
Response: Thank you for your comments. We have revised it to defect detection inside the tunnel, which is also common detection scene with GPR equipment.
Reviewer 3 Report
accept
Author Response
Thank you for the helpful comments. We carefully revised the manuscript, especially the English language. All changes in the updated manuscript are indicated with red highlighting.